

# Accounting for empirical global soil organic characteristics and moisture heterogeneities in soil organic decomposition scheme of land surface models.

Elodie Salmon[1,2], Bertrand Guenet[1], Agnès Ducharne[3]

[1]Laboratoire de Géologie, École normale supérieure, CNRS, PSL Univ., IPSL, Paris, France.
[2]Laboratoire des Sciences du Climat et de l'Environnement, CEA-CNRS-UVSQ, Gif-sur-Yvette, France
[3]Sorbonne Université, CNRS, EPHE,IPSL, UMR 7619 METIS (Milieux environnementaux, transferts et interaction dans les hydrosystèmes et les sols), 75005,Paris, France.

*Correspondence to*: Elodie Salmon (elodie.salmon@lsce.ipsl.fr), Bertrand Guenet (guenet@geologie.ens.fr)

**Abstract.**

Below ground soil organic carbon (OC) decomposition is governed by several biophysical drivers, causing difficulties to accurately capture the spatial patterns of soil OC stock and of $CO_2$ flux in Earth System models (ESMs). These biophysical drivers influence soil OC decomposition due to the respiration of heterotrophic organisms. Formulation in global scale process-based models of these processes consist of functions that modify the soil OC decay rate and therefore the soil heterotrophic respiration (HR) which modify global soil OC stock estimated by models. Current soil HR modifiers employed in models are a single relationship between soil moisture and the rate of decomposition that are employed for all the ecosystem types. Observational database meta-analysis relationships of SOIL MOISTURE and soil HR has been established considering observed soil physical properties. These relationships serve to define an empirical model that consists of a collection of different relationships based on soil organic carbon content, clay fraction and bulk density in order to uniquely substitute SOIL MOISTURE control on soil HR with a function modifier that reflects soil HR spatial heterogeneity.

In the present study, this empirical model has been embedded in the land surface model Organising Carbon and Hydrology In Dynamic Ecosystems (ORCHIDEE). The effect of the multivariate approach on simulation results has been assessed on soil OC stock and soil HR estimations at global scale. Results show that global soil OC stocks are nearly doubled in the modified model version, which is closer to observations-based products compared with the initial version, while $CO_2$ emissions, due to soil HR, are unchanged. The latitudinal soil OC distribution is maintained, displaying as much soil OC stock in tropical regions as under higher latitudes. This study demonstrates the significance of secondary drivers in the relationship between SOIL MOISTURE and the soil HR response to enable accounting for soil OC stock and $CO_2$ fluxes heterogeneous spatial pattern.



# 1 Introduction

The soil organic carbon stock changes are the result of the equilibrium between the organic carbon (OC) incorporated in ecosystems that is conveyed to the ground, lateral transport of soil OC through for instance, erosion and mineralization of soil OC into carbon dioxide which by returning to the atmosphere influences climate conditions. While the input of OC is yielded by way of photosynthesis and fluctuates depending on land-use and climate, soil carbon decay is predominantly issued from the respiration of heterotrophic organisms subjected to soil biophysical conditions. Among the environmental drivers influencing microbial activities, soil temperature and moisture prevails leading to the use of control functions on the rate of soil heterotrophic respiration (HR) in global scale process-based models. These models serve to simulate global soil OC stocks consistently with empirical data for the historical period and as a result project future soil carbon storage and emissions in order to improve our understanding of global soil OC stocks with above ground OC (biomass and litter) and climate (Todd-Brown et al., 2013; Varney et al., 2022).

Hence, the diversity of HR modifiers is reflected in the range of global soil OC stocks and HR fluxes estimated by ESMs (Ito et al., 2020; Varney et al., 2022, Guenet et al., 2024) which variability has been shown to be more than 43% of the average simulated global soil OC stock and 21% of the average simulated global HR across 15 ESMs. Ito et al. (2020) pinpoints that this inconsistency between models is issued from carbon decomposition rates divergence among ESMs which is the expression of processes and parameters implemented in the schemes of decomposition and in particular on HR modifiers in response to the influence of soil temperature and moisture.

Guenet et al., (2024) showed that precipitation is a key driver of the HR ESMs' residues suggesting that a better representation of the soil moisture effect on decomposition may be a good lead to improve HR representation in ESMs. Furthermore, the uncertainty of HR data-driven estimates is also widely dependent on the soil moisture product employed to constrain the database. Yao et al. (2020) found that the main drivers of interannual variability of HR are soil moisture and precipitation.

While the temperature-respiration relationships employed in ESMs are generally based on Q10 approach (Varney et al., 2022), a larger diversity of soil moisture-respiration relationships are applied. Varney et al., (2022) investigate variability in soil OC stocks estimate from ESMs involved in the Coupled Model Intercomparison Project (CMIP), CMIP5 and CMIP6, and distinguished two types of temperature and moisture schemes: (1) an increase relationship using Arrhenius law for temperature and a monotonically increase function with increasing soil moisture and (2) a hill-shape relationship such as Q10 formulation for the temperature dependence and hill function that increase to an optimum moisture level then decrease. They demonstrate that the soil OC stock to HR ratio sensitivity to temperature is more consistent among models than the sensitivity to soil moisture. Moreover, Falloon et al. (2011) appraised soil carbon changes responsiveness to control moisture- respiration functions embedding twelve representative climate models' functions in the RothC model. These functions have various shapes but provide the lowest of rate modifier values at the lowest soil moisture content and, for half of the functions, a rate modifier that is maximum at saturation. For the other half, a maximum rate near or higher than 0.5 is



given at the saturation fraction. They aimed at defining the responsive range of soil moisture on soil carbon changes and showed that soil carbon changes in a range of -2 to 3% between 1860 and 2100.

Falloon et al. (2011) also demonstrate that, in models, soil carbon changes are more responsive to climate forcing but are independent of the initial global soil carbon stock. This is consistent with the work of Ito et al. (2020) that established a linear relationship for all the ESMs of the study between changes in cumulative carbon input by litterfall and the carbon output issued from the HR. Conversely, the latest regional data-driven HR estimate by Ciais et al. (2020) suggests otherwise. Indeed, the HR to net primary productivity (NPP) ratios of 9 large regions are lower than 1 and the average ratios of each region range between 0.37 and 0.85 suggesting a non-linear relationship between the C input supply by litterfall and CO2 emissions resulting from HR, driven by carbon lateral transfer to aquatic ecosystems.

Other conditions such as soil pore space, bulk density and texture are neglected in process-based models regardless of their influence on metabolic activities of aerobic organisms (Moyano et al. 2013). Moyano et al. (2012) provide an analysis of the soil moisture response on HR of a wide range of mineral soil types and organic-richer soils. From this database, they provided a multivariable model dependent on soil moisture and soil characteristics namely soil organic carbon content, clay fraction and bulk density. Thus, this empirical model is not an average representation of the relationship between soil moisture and the HR but an ensemble of these relationships for various types of soils enabling to considered spatial heterogeneity. Regardless of its meaningful quality, this empirically-based model has never been employed in a process-based model.

In the present study, we examined the influence of soil characteristics on the soil carbon-storage and HR spatial pattern. To achieve that, Moyano et al. (2012) empirical model is employed for accounting for soil decomposition spatial heterogeneity including primary and secondary control drivers in the land ecosystem model ORCHIDEE (Krinner et al., 2005; Boucher et al., 2020; Tafasca et al., 2020). To prevent model complexity from hindering simulation results interpretation, we employed the bucket C model scheme version of ORCHIDEE which accounts for bulk litter and soil OC contents and does not account for higher latitude OC rich soils physical processes.

## 2 Methods

### 2.1 Model description

Moyano et al. (2012) established empirical models accounting for the dependence of the Proportional Response of Soil Respiration (PRSR) with soil characteristics. We embedded in the land surface model ORCHIDEE v2.2 revision 8416, named hereafter ORCHIDEE, the simplified version for mineral soil of these empirical models that rely on soil carbon content, clay content and bulk density and the version for organic-richer soils. ORCHIDEE is a land surface model in which CO2, water, and heat exchanged between the surface and the atmosphere are computed at a half-hourly time-step, and the carbon pools fluxes are computed at a daily time-step (Krinner et al., 2005). Total soil OC stocks are considered as a bulk





amount, not vertically discretized, preventing permafrost OC dynamics representation and does not presuppose a SOC maximum depth; instead, directly compute the SOC content per unit of surface area (m2). The soil carbon module originates from the scheme in CENTURY (Parton et al., 1987) that is composed of three pools named active, slow and passive to apprise on their respective residence times. The litter and soil OC dynamics for each pool (p) is defined by :

$$\frac{d[C]_p}{dt} = I_p - k_p \cdot [C]_p \cdot f(\theta) \cdot f(\tau) \cdot f(\gamma) \qquad (1)$$

where $[C]_p$ is the carbon content (g C / m$^2$) in the litter or in the soil pools, p specifies the pool nature among aboveground metabolic litter, belowground metabolic litter, aboveground structural litter, belowground structural litter, active, slow and passive soil carbon pools, I is the input of C (g C / m$^2$) and k is the decomposition rate. Modifying the OC decomposition, $f(\theta)$, $f(\tau)$, and $f(\gamma)$ are the moisture function, the temperature function, and the texture function respectively, accounting for

environmental constraints independently from the other modules.

On one hand, in the reference version of ORCHIDEE employed in the present study, the moisture function θ is defined by eq. 2

$$f(\theta) = max(0.25, min(1, -1.1 \times \theta^2 + 2.4 \times \theta - 0.29)) \qquad (2)$$

with θ being the soil moisture in m3/m3 and ranges between 0.25 and 1.

On the other hand, in a modified model version named hereafter ORCHIDEE-M, the moisture function f(θ) has been replaced by the PRSR empirical models from Moyano et al., (2012) to modify the litter and soil organic carbon dynamics described in Equations 1 and 2. The PRSR empirical models are constructed for soil moisture in fraction of saturation (θ$_s$) intervals of 0.01 ranging between 0 and 1 using the relationships in Eq3 for mineral soil and in Eq4 for organic-richer soil:

$$PR_{SR}(\theta_s) = \beta_1 \theta_s + \beta_2 \theta_s^2 + \beta_3 \theta_s^3 + \beta_4 \, clay + \beta_5 \, clay \, \theta_s + \beta_6 \, [C]_{TSOC} + intercept \qquad (3)$$

$$PR_{SR}(\theta_s) = \beta_1 \theta_s + \beta_2 \theta_s^2 + \beta_3 \theta_s^3 + intercept \qquad (4)$$

where β are the empirical model parameters defined as in Table 1, clay is the clay fraction values, $[C]_{TSOC}$ is the total soil organic carbon content in g C / g of soil and the intercept of the empirical model for a null PRSR. The soil OC content is estimated in g C / m$^2$ of soil in ORCHIDEE and is converted to g C / g of soil to be used in the PRSR empirical model using soil bulk density (g / m$^3$) and assuming a soil height of 0.2 meter. The clay fraction is defined using Zobler (1986), soil bulk

density values are gridded dependent, established from the Harmonized World Soil Database (HWSD) soil map (Fischer et al., 2008).

In order to maintain integrity of the empirical model, environmental constraints, i.e. clay fraction, soil OC and the bulk density, are retained within ranges defined by Moyano et al. (2012) based on the soil samples used to fit linear regression models. Thus, the Moyano et al., (2012) function is forced by clay fraction ranging between 0.03 and 0.58, bulk

density between 0.8 and 1.5 g/cm$^3$, and soil OC between 0.01 and 0.35 g C / g soil. For a soil OC lower than 0.06 g C / g soil, the mineral soil model, i.e. Eq. 3, is employed; otherwise, for higher soil OC content, the organic-richer soil model, i.e. Eq. 4, is applied.



**Table 1: Model parameters values of Moyano et al. (2012)'s simplified version for mineral soil and organic-richer soil (SOC: Soil Organic Content).**

| Parameters | $\beta_1$ | $\beta_2$ | $\beta_3$ | $\beta_4$ | $\beta_5$ | $\beta_6$ | intercept |
|---|---|---|---|---|---|---|---|
| mineral soil model for 0.01>SOC>0.06g/g | -0.26 | 0.32 | -0.15 | 0.08 | -0.09 | 0.57 | 1.059 |
| organic soil model for SOC >0.06g/g | -0.67 | 1.08 | -0.57 | -- | -- | -- | 1.134 |


The moisture function is then defined by:

$$MPR_{SR}(\theta_s)_n = SR_{ini} \frac{PR_{SR}(\theta_s)_n \times MPR_{SR}(\theta_s)_{n-1}}{MPR_{SR}(\theta_s)_{max}} \qquad (5)$$

where $SR_{ini}$ is the initial respiration value and is assumed to be 1.0, n is the soil moisture content interval number employed to estimate the $PR_{SR}$ using equation 2. For the first interval number (n=1), the value of $MPR_{SR}(\theta_s)_{n-1=0}$ is 1. The soil

moisture control function ($MPR_{SR}$) is scaled to range between 0 and 1 by subtracting all values with the smallest value then normalizing with the highest one. At each timestep, the $MPR_{SR}$ values employed to constrain soil OC decomposition by HR is determined by the soil moisture content estimated by the model (Fig. 1).

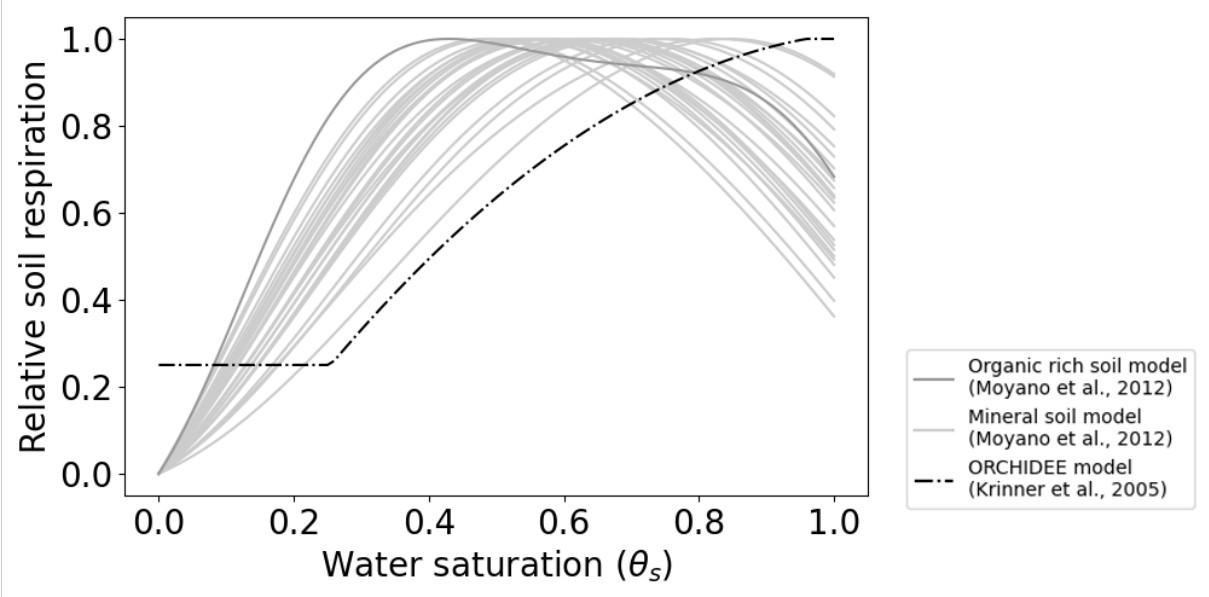

**Figure 1: Representation of the control moisture function in ORCHIDEE (dash-dotted line, Krinner et al., 2005) and of the**
**ensemble of control moisture function embedded in the modified ORCHIDEE-M version for the mineral soil (light grey solid lines) using clay fractions ranging between 0.27 and 0.34, soil OC ranging between 0.01 and 0.06 g C / g soil and for organic-richer soil (dark grey solid line) with soil OC higher than 0.06 g C / g soil.**



## 2.2 Simulation protocol

145       The standard and modified ORCHIDEE versions, named respectively hereafter ORCHIDEE and ORCHIDEE-M, were run at global scale with a 0.5 x 0.5 degree resolution employing historical and present climate and atmospheric $CO_2$ forcing. The historical climate data are a 6-hourly forcing product based on the second Japanese global atmospheric reanalysis, named the Japanese 55-year Reanalysis (JRA; Kobayashi et al., 2015) corrected by the Climate Research Unit (CRU; Mitchell et al., 2004; Harris et al., 2014). First, the ecosystem equilibrium was reached for both model versions by

repeating in a loop the first decade of the historical climate forcing (1901–1910), and employing atmospheric $CO_2$ pre-industrial values corresponding to year 1850. Second, both models run during 49 years prescribing climate data from random years between 1901 and 1910 but with 1851-1900 times series for land-use change and $CO_2$ data. Then, models were run for the period 1901-2010 by historical climate, land use maps of the LUHv2h dataset (Chini et al., 2021; used in the "Trends and drivers of the regional scale terrestrial sources and sinks of carbon dioxide" (TRENDYv11 – Sitch et al. 2024) project for the

Global Carbon Budget, Friedlingstein et al., 2022) and $CO_2$ forcing data.

## 2.3 Observation dataset

       To evaluate the model, a global-scale benchmarking of simulated carbon stock was performed using three soil datasets estimating soil OC stock from 0 to 1m namely the Global Soil Dataset for use in Earth System Models (GSDE-

Shangguan et al., 2014), Harmonized World Soil Database version 2.0 (HWSD v2.0 - FAO & IIASA, 2023) and the global gridded soil system SoilGrids (Hengl et al., 2017). Global HR is benchmarked against four HR estimations which are the upscaling of in situ measurements of Hashimito et al. (2015), the higher resolution in situ based-estimate of Warner et al. (2019), the top-down global estimate of Koning et al. (2019) and the machine learning data-driven estimate of Yao et al. (2021).

## 3 Results

165       The control moisture- respiration function of ORCHIDEE has been replaced by the ensemble of empirical multivariate soil moisture response on HR (Moyano et al., 2012). Indeed, ORCHIDEE control moisture function (CMF) is a function (Fig. 1) reaching saturation for water saturated soil (soil moisture =1, CMF=1) which is inconsistent with field observations and laboratory experiments (Moyano et al. 2013; Tang and Riley, 2019). These studies showed that microbial

respiration maximizes at average volumetric soil moisture values ranging between 0.5 to 0.8. At higher volumetric soil moisture, microbial respiration is significantly reduced owing to a limitation in interstitial oxygen content.

       The bucket C model scheme version of ORCHIDEE is employed here to ease the interpretation of the results. In this scheme, the soil OC content is not vertically discretized and is not accounted for in the energy module meaning that organic-rich soils insulation is not simulated (Gaillard et al., 2025). However, this setup enables distinguishing the influence





of both control moisture functions based solely on the OC decomposition scheme. At each time step and for each model grid cell and soil tile, the model is constrained with the soil clay fraction and the bulk density from observation databases described in section 2.1 model description. The bulk density serves to convert soil OC content between kg C / m2 soil and g C / g soil. The soil OC content and the volumetric soil moisture is estimated by the model which enables to define a unique value of the control moisture - respiration functions for the litter and for the soil. The following results analysis is performed

using yearly average simulation results between years 1990 and 2000.

## 3.1 Control moisture function (CMF) analysis

Differences in the CMF determined in both model versions, ORCHIDEE and ORCHIDEE-M for the litter (on the left side) and the soil (on the right side) are displayed in Fig. 2. The bar plots (A to D) show the dispersal of grid cells number for various CMF ranges of values. Values of the litter and soil moisture function are lower in the ORCHIDEE-M

version. In the standard model version, for the litter and the soil, 37% and 54% of the grid cells respectively have CMF values higher than 0.9 meaning that litter and soil moisture content has no significant influence on the HR. Values of the CMF ranging between 0.5 and 0.9 are used in 47% and 43% of the grid cells for the litter and the soil respectively, implicating little control on HR. In the ORCHIDEE-M version, the largest number of grid cells corresponding to nearly 70% of the grid cells, have CMF values ranging between 0.2-0.6 for the litter and 0.3-0.7 for the soil. Fewer grid cells have soil

CMF values higher than 0.8 for the ORCHIDEE-M version than ORCHIDEE version, i.e., 10% and 72% of grid cells for the ORCHIDEE-M and ORCHIDEE versions respectively, promoting, in the standard model version, soil oxic decomposition rather than soil OC accumulation.

In Fig. 2E and 2F maps, negative values indicate that CMF values are higher in the standard version, enabling a faster soil OC decomposition. In deserts and dryer soils CMF values in the ORCHIDEE-M version ranges between 0.2 and 0.45

establishing a stronger control than in other regions. In tropical wetland areas such as the Indonesia, the Amazon and the Congo basins, both versions prescribe high CMF values, higher than 0.9 whereas elsewhere between 20°N and 20°S latitude CMF values are mixed with in the most tropical humid regions like central of south America and south east Asia, values higher than 0.6 and in the driest tropical areas, value ranging between 0.2 and 0.6 for the ORCHIDEE-M version. Northern than 40°N, the CMF values are mainly higher than 0.4, serving to accumulate a higher soil OC content using the

ORCHIDEE-M than using the ORCHIDEE version.

Of the total number of continental surface grid cells, 46% have CMF values that only slightly change which display anomalies equal to zero down to -0.1. Most of the other grid cells, i.e. 16% have CMF anomaly values ranging between -0.1 and -0.5 and remaining 1.2% are between -0.5 and -0.9. By regional band, first in the tropical areas between 30°S and 30°N, 20185 grid cells, i.e. 24% of the tropical band cells, have CMF anomaly values between -0.1 and -0.5. Others, i.e. 76% of the

tropical band cells, have anomaly values ranging between 0 and -0.1 for which CMF values are similar in both model versions. In the temperate latitudinal bands, between 30°N and 60°N, 3316 grid cells, i.e. 8% of continental surface cells in the northern temperate band, have CMF anomaly values lower than -0.5 and are mostly located in boreal areas. Near zero



anomaly ranging between 0 and -0.1 revealing a mild diminution in soil OC decomposition rate is obtained for 48% of continental surface cells in the northern temperate band. Remaining 44% of continental surface cells in the northern temperate band have CMF anomaly values ranging between -0.1 and -0.5. In the southern temperate band (30°S and 60°S), 95% of the grid cells are higher than -0.1 and the remaining 5% of continental surface cells display CMF anomaly values ranging between -0.1 and -0.7. In the arctic regions north of 60°N, 60% of continental surface cells of the northern high latitude band, have CMF anomaly values higher than -0.1, 23% range between -0.1 and -0.5 and 17% that are lower than -0.5.





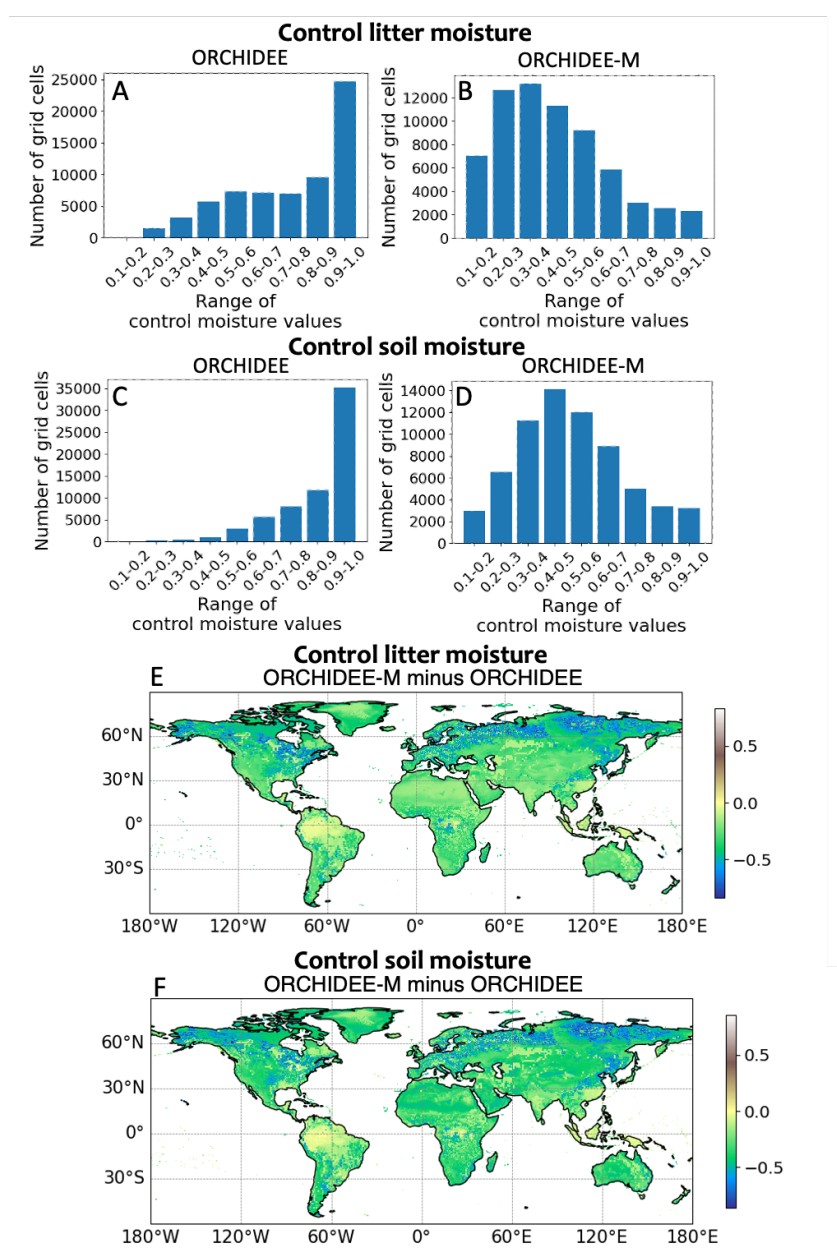


**Figure 2: At the top, bar charts of the number of grid cells per control moisture function value ranges for the litter (A) and the soil (C) in the ORCHIDEE version, for the litter (B) and the soil (D) in the ORCHIDEE-M version. The bottom rows are maps of values difference between the control moisture function in the ORCHIDEE and ORCHIDEE-M versions for litter (E) and for soil (F).**

To underline secondary drivers' influence on OC decomposition, Fig. 3 displays the relationship between the OC content and the moisture content whereas the coloured bars show the repartition depending on grid cell latitudes, CMF values and clay fraction. The OC and the moisture content relationship exhibit a higher OC content for moisture content




lower than 0.4 for the litter and lower than 0.6 for the soil in the modified ORCHIDEE-M version. This is because, in the
ORCHIDEE standard version, the CMF is defined independently from the soil clay and OC content whereas in the
ORCHIDEE-M version these soil characteristics are accounted for. Colors, in the Fig. 3 first row, indicate latitudinal
location of the grid cells depending on their OC and moisture content. Grid cells with the highest OC content are located in
the southern hemisphere (brown and yellow coloured data points in the upper line of Fig. 3) for both model versions. Grid
cells located in the northern hemisphere in blue and green coloured data points have higher OC content in drier soil in
ORCHIDEE-M than ORCHIDEE. Figures in the second row of Fig. 3, reveal the CMF values used in each grid cell.
Similarly than in Fig. 2, the litter and soil CMF values in the ORCHIDEE version have mainly a value of 1 and few of the
grid cells have CMF values around 0.4 for dryer litter and soil moisture content, lower than 0.25. In the modified
ORCHIDEE-M version, the CMF values have an increasing value from lower values for lower litter and soil moisture
content and higher value up to 1 for more saturated litter and soil moisture content.  The clay fraction does not display an
obvious pattern, still most of the grid cells with higher clay fraction also have higher OC content and higher moisture content
whereas grid cells with a clay fraction lower than one, have the lowest moisture content around 0.1.

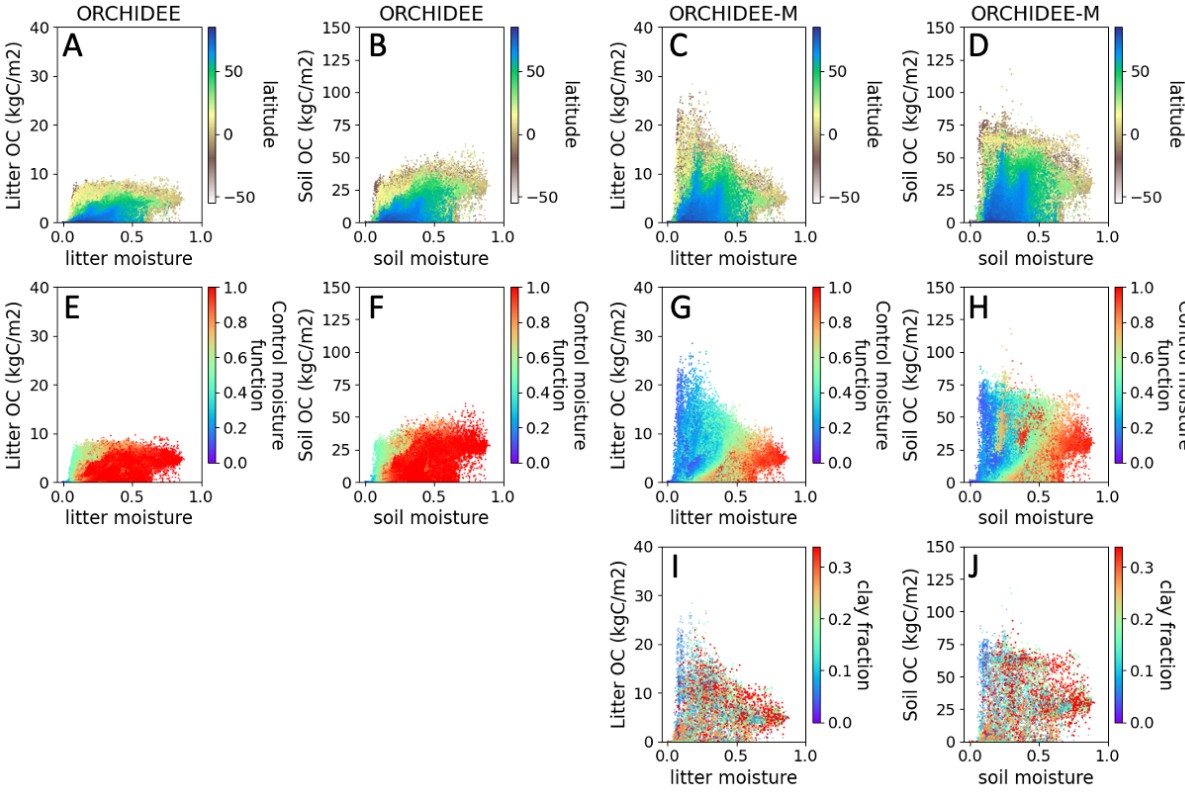

**Figure 3: Influence of latitude location (first row, A to D), CMF value (second row, E to H) and clay fraction (third row, I and J) on the relationship of litter and soil OC and moisture content for the standard ORCHIDEE model version (left columns) and the modified ORCHIDEE-M model version (right columns).**



## 3.2 Litter and Soil OC stock assessment


In both model versions, the OC stock rate depends on the OC decomposition rate, for the litter, on the material originating from the net primary productivity and for the soil OC stock, on the input of the litter OC content. The litter and soil OC stock accumulated globally in both model versions have been evaluated against three soil OC databases. Because the litter and soil OC contents are simulated as bulk amounts per unit of surface area (kg C / m$^2$), the vertical dimension is not

defined in the models, preventing simulated OC stock to be directly compared against observation databases. Therefore, we assumed in the present study that the total litter and soil OC content accumulated up to 1 m soil depth and evaluated the simulation results against the following observation databases: GSDE, HWSD v2.0 and SoilGrids at 1m.

The first one is the GSDE estimation that is based on soil attribute data and soil maps from the Soil Map of the World and various regional and national soil databases (Shangguan et al., 2014). The second one is the HWSD v2.0 estimation which is

an improvement of the previous version 1.2 built from soil attribute data and soil profiles (FAO & IIASA, 2023). The third one, SoilGrids, is the result of a spatial prediction of global organic carbon stored in the first 1 meter of soil, using a compilation of soil organic carbon profiles and samples. Thus, SoilGrids global soil OC stock estimation of the top 1 meter of soil provides the slightly highest soil OC stock estimate at 2331 PgC, whereas GSDE and HWSD v2.0 correspond to 1720 and 2049 PgC, respectively.


Considering in the ORCHIDEE-M formulation, emergent environmental soil characteristics and empirical relationship of soil moisture with heterotrophic respiration, enable a total litter OC content increase of 89 PgC in the litter pools and by 424 PgC in the soil pools (Fig. 4A). Simulated total ORCHIDEE-M OC estimate of 1265 PgC is closer to OC estimated by GSDE than HWSD v2.0 or Soilgrids, corresponding to 54 %, 62% and 74 % of SoilGrids, HWDS and GSDE estimates, respectively. The latitudinal profiles displayed in Fig. 4B, shows for the 40-80°N latitudinal band that the OC

density of both models, i.e. for ORCHIDEE-M and ORCHIDEE of respectively, 391 and 207 kgC / m$^2$, are at least half of the estimations by the databases, i.e. for Soilgrids, HWSD and GSDE of respectively, 729, 996 and 850 kgC / m$^2$. Under these latitudes, SoilGrids, HWDS and GSDE OC density estimates are larger because they account for organic rich soil such as those formed under peatland ecosystems and soils that belong to the Yedoma formation. In the model version employed in the present study, interactions between soil OC content, soil temperature and the hydrology in the permafrost areas are not

accounted for involving colder soil temperature in the highest latitudes. These specific ecosystems are not represented in both model versions which explain the soil OC discrepancy between ORCHIDEE, ORCHIDEE-M and the three databases. In the northern 20°-40° latitudinal band, estimates for the ORCHIDEE-M, ORCHIDEE, are in the same order of magnitude with a higher value for the ORCHIDEE-M soil OC density of 89 kgC / m$^2$ than for the ORCHIDEE version of 54 kgC / m$^2$ and a significantly higher estimate for SoilGrids, HWD and GSDE of respectively, 170, 142 and 120 kgC / m$^2$. In the

tropical area between 20°N and 20°S and in the southern hemisphere, the ORCHIDEE-M soil OC density of 230 kgC / m2 is similar than the estimates by Soilgrids, HWD and GSDE of respectively, 227, 237 and 188 kgC / m$^2$. In the same region, the ORCHIDEE soil OC estimate of 148 kgC / m$^2$ is underestimated compared to the databases and of the OC density estimate

from the ORCHIDEE-M version. Both versions of the model have been unable to capture the sharper peak of soil OC density in the equatorial band likely originating from tropical wetland ecosystems. In the absence of ecosystems capable of
forming organic rich soil, there is a compensation bias from other ecosystems for the tropical and southern hemisphere latitudinal regions.

**Figure 4 : Total litter and soil OC stocks, on the left panel (A) and latitudinal distribution (B) of the litter and soil OC stocks, on the right panel of the standard model version (ORCHIDEE) and the modified model version (ORCHIDEE-M) and for two**
**observation databases in orange.**

        The mean annual soil OC density maps for the period 1990 to 2000 and for both model versions are shown in Fig. 5. The increase in the ORCHIDE-M version of the soil OC density values is located in areas having the highest soil OC density, around 10 (5-15) kg/m$^2$ in the ORCHIDEE version. The other maps in Fig. 5- C to -H display the difference of annual mean soil OC density simulated by both model versions with the three databases. The standard ORCHIDEE model
largely underestimates the soil OC density compared to the three databases. Comparisons of GSDE and HWSD with the ORCHIDEE estimate display a larger discrepancy in areas colored in dark blue and located northern than 50°N, in the



western Siberia, the Ob basin, Scandinavia, Hudson Bay and in Alaska. These areas correspond to the northern sphagnum dominated peatlands that are known to store nearly a third of the total soil OC stock. The modified ORCHIDEE-M version overestimates the soil OC density of around 20 kg/m$^2$ in the tropical areas between 35°N and 35°S (brown coloured areas) in the comparison with the three databases. It also underestimates the other areas more particularly the organic-rich soil accounted for in the GSDDE and HWSD data in the high latitudes northern than 50°N as the standard version.

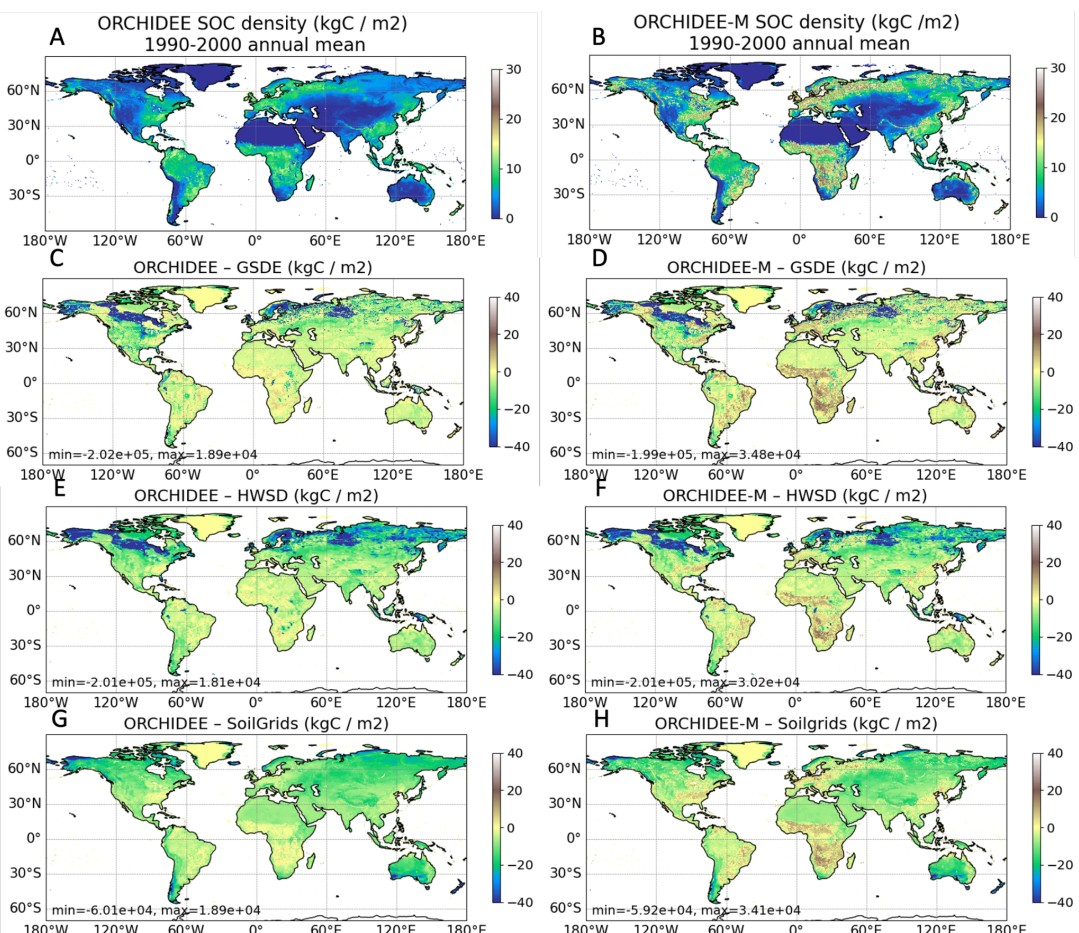

**Figure 5: Annual mean soil OC density map for the period 1990-2000 of the model ORCHIDEE and ORCHIDEE-M (A-B) and differences (C to H) of simulated annual mean soil OC density and of the three databases Soilgrids, HWSD and GSDE in kgC /m$^2$ (in the three-bottom row).**

An evaluation of the ecosystem contribution (Table 2) considered in the models using plant functional types (PFTs) provides insight on implications involved by the formulation changes between both model versions. Both litter and soil OC accumulated per PFTs increase for all PFT types and the percent increase of litter and soil OC content per PFT follow similar trends. The highest soil OC stock increase that occurs northern than 40°N (Fig. 4B), originates from boreal forest





(BNLE, BBLS and BNLS) and grassland (BC3G) PFTs and has a OC increase, in litter and soil, of more than 84%C of OC differences. In temperate areas, litter and soil OC stock increased under the forest (TeNLE, TeBLE and TeBLS) between 55 and 78% of OC and by more than 72% under grasslands (TeC3G) and crops (C3Agri and C4Agri). In tropical regions, litter
and soil OC stock has intensively increased by more than 71% under tropical broad-leaved raingreen forest (TrBLR) and grassland dominated by C4 plants (C4 Grass).

**Table 2: Simulated litter and soil organic carbon content from both model versions ORCHIDEE (ORC) and ORCHIDEE-M (ORC-M) and for the ecosystems considered in the model. The organic carbon content differences of ORCHIDEE minus**
**ORCHIDEE-M version is provided in petagram and in percent of the ORCHIDEE OC content estimated for each plant functional type (PFT) which are TrBLE for Tropical Broad-leaved Evergreen, TrBLR for Tropical Broad-leaved Raingreen, TeNLE for Temperate Needle-leaf Evergreen, TeBLE for Temperate Broad-leaved Evergreen, TeBLS for Temperate Broad-leaved Summegreen, BNLE for Boreal Needle-leaf Evergreen, BBLS for Boreal Broad-leaf Summegreen, BNLS for Boreal Needle-leaf Summergreen, TeC3G for Temperate C3 Grass, C4G for C4 dominated grassland, C3Agri for C3 Agricultural plants, C4Agri for**
**C4 Agricultural plants, TrC3G for Tropical C3 dominated Grassland and BC3G for Boreal C3 dominated Grassland. Fractions of PFTs indicate the percentage of total area of each PFT of the total land surface.**

| | | Litter | | | | Soil | | | |
|---|---|---|---|---|---|---|---|---|---|
| Plant Functional Types | Fraction | ORC | ORC-M | Differences | | ORC | ORC-M | Differences | |
| | % | Pg | Pg | Pg | %C | Pg | Pg | Pg | %C |
| TrBLE | 9.4% | 10.6 | 13.8 | 3.3 | 31% | 59.8 | 75.3 | 15.5 | 26% |
| TrBLR | 6.0% | 5.2 | 9.1 | 3.9 | 75% | 27.5 | 47.0 | 19.5 | 71% |
| TrC3G | 3.3% | 0.9 | 1.4 | 0.5 | 57% | 5.0 | 7.8 | 2.8 | 57% |
| C4 Grass | 3.2% | 16.1 | 31.3 | 15.2 | 94% | 75.5 | 138.3 | 62.8 | 83% |
| TeNLE | 3.5% | 1.6 | 2.6 | 1.0 | 61% | 9.1 | 14.2 | 5.1 | 55% |
| TeBLE | 4.8% | 4.0 | 7.0 | 3.0 | 75% | 23.0 | 38.0 | 15.1 | 66% |





| | | | | | | | | | |
|---|---|---|---|---|---|---|---|---|---|
| TeBLS | 3.8% | 3.6 | 6.4 | 2.8 | 78% | 20.4 | 33.7 | 13.3 | 65% |
| TeC3G | 2.7% | 4.6 | 8.2 | 3.6 | 78% | 25.3 | 43.4 | 18.1 | 72% |
| C3Agri | 6.8% | 8.1 | 14.3 | 6.2 | 77% | 35.9 | 62.6 | 26.6 | 74% |
| C4Agri | 9.0% | 3.6 | 6.5 | 2.9 | 80% | 12.1 | 20.9 | 8.8 | 73% |
| BNLE | 9.1% | 3.8 | 7.2 | 3.4 | 88% | 19.1 | 35.1 | 16.0 | 84% |
| BBLS | 2.2% | 3.5 | 6.7 | 3.2 | 89% | 19.1 | 35.3 | 16.1 | 84% |
| BNLS | 4.1% | 1.4 | 2.8 | 1.5 | 108% | 8.6 | 18.5 | 9.9 | 115% |
| BC3G | 11.4% | 6.0 | 11.3 | 5.3 | 89% | 32.1 | 60.6 | 28.5 | 89% |

### 3.3 Evaluation of CO2 fluxes from heterotrophic respiration (HR)

Three estimations of $CO_2$ emissions originating from HR serve to evaluate the $CO_2$ simulation results. Simulated
$CO_2$ emissions by both model versions are higher i.e. 52.7 and 52.3 Pg C / yr for ORCHIDEE and ORCHIDEE-M versions
respectively, than the $CO_2$ fluxes estimated using in-situ data i.e. 51.5 and 50.0 Pg C / yr by Hashimoto et al. (2015) and
Warner et al., (2019), and using top-down inversions model i.e. 43.4 Pg C by Konings et al., (2019). Thus, global HR flux
estimated by the ORCHIDEE-M version has only slightly changed in comparison to the estimation obtained using the
standard version ORCHIDEE. Although, the latitudinal distribution of the $CO_2$ flux (Fig. 6B) shows that in tropical and
subtropical regions, between 40°N and 40°S, both model versions overestimate HR by about 45% of the observation mean.
Northern than 40° north, simulated HR fluxes are lower by 14 and 15%, respectively for ORCHIDEE and ORCHIDEE-M
than the mean estimations of the observation databases. Similarity of $CO_2$ fluxes latitudinal profiles for both model versions
in the tropical and subtropical regions are consistent with the CMF values that only slightly changed between the two model
versions. Indeed, these regions have dry desert areas, depleted in soil OC content soil moisture content hence limiting $CO_2$
fluxes and other areas for which soil moisture content is higher than 0.5 leading in both cases to CMF values higher than 0.6.
The higher the CMF value is the least it influences soil organic decomposition rate (equation 1).







**Figure 6: A- Global HR CO₂ fluxes estimations (on the left panel) obtained using the ORCHIDEE and ORCHIDEE-M models in blue and from observation databases in orange. B-Distribution of the simulated (red and blue lines) and observations (green solid, dashed and dot dashed lines) HR CO₂ fluxes along the latitudinal profile (on the right panel).**

Maps of the mean annual HR for the period 1990 to 2000 and for both model versions are merely identical as shown in Fig. 7A and B. The largest HR CO₂ fluxes are simulated in the tropical and subtropical latitudinal bands located between 30°N and 30°S. Comparison with the three databases from Hashimoto et al. (2015), Warner et al., (2019), and Konings et al., (2019) in Fig. 7C to H, reveal that these simulated HR CO₂ fluxes in the 30°N-30°S band are overestimated by more than 1 kg / m² / y. Other drier areas such as central Australia, Sahara, the Arabian Peninsula and the south western of the United States of America have slightly underestimated simulated HR CO₂ fluxes compared to the Hashimoto et al. (2015), Warner et al., (2019) estimates.





**Figure 7: Annual mean HR CO₂ fluxes map for the period 1990-2000 of the model ORCHIDEE and ORCHIDEE-M (A-B) and**
**differences (C to H) of simulated annual mean HR CO₂ fluxes and of the three databases of Hashimoto et al. (2015), Warner et al., (2019), and Konings et al., (2019) in kilograms per square meter per year (in the three-bottom row)**

Repartitions of simulated HR CO₂ fluxes depending on vegetation land cover are displayed in Table 3. HR estimation differences correspond to differences for each PFT between the modified version, ORCHIDEE-M, and the standard model version, ORCHIDEE. Differences are minor and negative values for all land cover types. Three biomes can

be distinguished in Table 3, the tropical (TrBLE, TrBLR, TrC3G, C4 Grass), temperate (TeNLE, TeBLE, TeBLS, TeC3G, C3Agri, C4Agri) and boreal (BNLE, BBLS, BNLS, BC3G) biomes that account, respectively, for 28, 28, an 23% of the total





land surface area and for which the HR $CO_2$ fluxes differences are respectively of -2, -6 and -4% of the ORCHIDEE estimates. Hence the largest dissimilarities between both simulated HR $CO_2$ fluxes of around -1.3% coincide with temperate and boreal biomes, more specifically with TeBLS, TeC3G, BNLE forest and BC3G grassland.

**Table 3: Simulated heterotrophic respiration $CO_2$ fluxes from both model versions ORCHIDEE and ORCHIDEE-M. The CO2 flux differences of ORCHIDEE minus ORCHIDEE-M version is provided in teragram (Tg) and in percent of the ORCHIDEE CO2 fluxes estimated for each plant functional type (PFT) which are TrBLE for Tropical Broad-leaved Evergreen, TrBLR for Tropical Broad-leaved Raingreen, TeNLE for Temperate Needle-leaf Evergreen, TeBLE for Temperate Broad-leaved Evergreen, TeBLS for Temperate Broad-leaved Summegreen, BNLE for Boreal Needle-leaf Evergreen, BBLS for Boreal Broad-leaf**
**Summegreen, BNLS for Boreal Needle-leaf Summergreen, TeC3G for Temperate C3 Grass, C4G for C4 dominated grassland, C3Agri for C3 Agricultural plants, C4Agri for C4 Agricultural plants, TrC3G for Tropical C3 dominated Grassland and BC3G for Boreal C3 dominated Grassland.**

| Plant Functional Types | ORCHIDEE | ORCHIDEE-M | Differences | |
|---|---|---|---|---|
| | Tg / yr | Tg / yr | Tg / yr | %C |
| TrBLE | 20.14 | 20.11 | -0.028 | -0.14% |
| TrBLR | 8.64 | 8.60 | -0.033 | -0.39% |
| TrC3G | 1.51 | 1.51 | 0.000 | 0.01% |
| C4 Grass | 21.05 | 20.82 | -0.230 | -1.09% |
| TeNLE | 1.87 | 1.86 | -0.014 | -0.72% |
| TeBLE | 3.45 | 3.44 | -0.013 | -0.38% |
| TeBLS | 4.13 | 4.08 | -0.051 | -1.22% |
| TeC3G | 4.81 | 4.75 | -0.058 | -1.21% |
| C3Agri | 8.28 | 8.20 | -0.083 | -1.00% |





| C4Agri | 4.06 | 4.01 | -0.045 | -1.12% |
| BNLE | 2.34 | 2.31 | -0.030 | -1.28% |
| BBLS | 2.34 | 2.31 | -0.026 | -1.12% |
| BNLS | 0.67 | 0.67 | -0.001 | -0.14% |
| BC3G | 3.19 | 3.15 | -0.038 | -1.19% |

## 4 Discussion

In land surface models (LSMs), CMFs are employed to limit the soil OC decomposition rate and govern soil OC stock and HR flux estimates (Varney et al., 2022). Multiple CMF shapes have been assessed in soil OC models (Falloon et al., 2011) and ESMs (Ito et al. 2020, Varney et al., 2022). For instance, in the study by Varney et al., (2022), they classified the CMFs employed in ESMs involved in CMIP5 and CMIP6 in two groups, the hill shape and the monotonically increasing functions. It will be expected from the hill shape functions to be more consistent with observations since they enable to

simulate a microbial optimum at the mid-range of soil moisture. At higher soil moisture content ($\theta_s > 0.5$), unlike the monotonically increased function for which the CMFs have limited or no effect on the HR response, is able to provide some constraints resulting in a better efficiency in soil OC accumulation. Nevertheless, Varney et al. (2022) found no consistent correlations between the CMF shapes and the soil OC stocks or HR $CO_2$ flux estimations in the ESMs involved in CMIP5 and CMIP6.

In addition to be composed of hill shape functions, the multivariate empirical model (Moyano et al., 2012) employed in the present study, accounts for spatial heterogeneities of soil characteristics, by way of an ensemble of relationships relying on the soil OC content, the soil bulk density and the clay fraction. This model results from a data-driven analysis of 310 estimations of the soil oxic decomposition response to soil moisture based on 90 soil samples from 42 sites which pledge for a better consistency of the model results with observations.

Results show that the total global carbon stock has doubled whereas the HR $CO_2$ flux is not significantly changed between the standard and the modified ORCHIDEE version employed in the present study. The soil OC stock rise varies between 3 and 63 Pg C for the different vegetation cover types and infers a positive bias at low latitudes whereas the HR CO2 fluxes are merely unchanged. The soil OC stock bias is independent of the vegetation cover and maintains the





overestimation of HR $CO_2$ flux at the same low latitudes suggesting the need to calibrate the models for the tropical band.

Our objective here was to build a new version of the ORCHIDEE model able to better represent the soil moisture effect on SOC decomposition but it is important to note that in such models, the SOC content is also largely controlled by NPP and it is also key to improve NPP representation in order to improve SOC dynamic in the model. NPP can be improved by changing some key parameters but another way to adjust NPP will be to employ a model version including carbon-nitrogen interactions as suggested by Varney et al. (2022) which improved simulated NPP estimates in the latest generation of models

involved in CMIP6. Even with the improved soil OC stock at medium and high latitudes, soil OC content is not reaching the soil OC content estimated by the databases, Soilgrids, GSDE and HWSD. It should be pointed out that wetland ecosystems such as northern and tropical peatlands, floodplains and mangroves are not explicitly represented in these versions of the model. Indeed, vegetation cover types are latitudinal dependent and account for northern (BC3G) and tropical (TrBLE, TrBLR) peatlands and mangrove phenology, however soil physico-chemical processes and characteristics of wetlands are

missing. Notwithstanding the lack of explicit wetland soil processes, the simulated soil OC stocks by the modified ORCHIDEE version reach 170 Pg at higher latitudinal bands 40°N-80°N and 300 Pg in tropical regions at 40°N-40°S. These amounts are to be compared to Jackson et al. (2017) estimates that assessed northern and tropical peatlands soil OC stock at 0-2m depth and for the same latitudinal bands to 269 and 104 Pg C, respectively. Thus, the empirical model enables not only to increase soil OC stock but also to improve the spatial distribution of soil OC stocks. Varney et al. (2022) also disclosed

that considering a vertical discretization of the soil OC budget enabled the ESMs involved in CMIP6 (Lawrence et al., 2019) to simulate the large northern high latitude soil OC stock. Certainly, the vertical C discretization schemes used in LSMs that are composed of an explicit representation of northern peatlands (Qiu et al., 2022) provide a vertical distribution of the soil OC which enables to account for the deeper soil OC storage. Some ongoing efforts to include N and soil vertical distribution in the main ORCHIDEE version are ongoing (Vuichard et al. 2018, Gaillard et al. 2025) but in this study we decided to work

on a more simplified scheme of soil OC decomposition to reduce the potential feedbacks and non-linearities to better understand the direct effect of changing CMF. Inclusion of the new CMF presented here would likely affect SOC and also N release, leading to indirect impact on NPP leading to a more complex response.

The relationship of simulated soil OC stock and the input and output fluxes from the soil known as NPP and HR CO2 emissions furnish a temporal effective global evaluation of the model results. Since the HR $CO_2$ emissions have not

changed, the median HR/NPP ratios of 0.8 are unchanged between both model versions and are consistent with the median estimation of 0.78 by Ciais et al. (2020). Regionally, HR to NPP ratios are in good agreement with Ciais et al. (2020) estimates for regions located in temperate and northern latitudes and overestimated in regions located in the low latitudes. However, the soil OC to HR ratio, sometimes define as a surrogate for turnover, of 13 years in the ORCHIDEE standard version, increased to 22 years in the ORCHIDEE-M modified version that is accordant with the recent mean estimate of 27

years (ranging to 54 - 16 years) defined from observation data by Varney et al. (2022) to assess CMIP models.



## 5 Conclusion

In the past 10 years, ESMs faculties to estimate soil OC and HR fluxes have been evaluated in several studies (Todd-Brown et al., 2013; Ito et al., 2020; Varney et al., 2022, Guenet et al., 2024) which showed (1) significant variability of soil OC and HR $CO_2$ fluxes estimates across models (2) that the observed spatial pattern of soil OC stocks have not been significantly improved and (3) a strong correlation between soil OC and NPP in models inconsistent with observations. These studies raised the concern on the reliability of ESMs to estimate future global carbon budget and provide guidance in assessing remedial solutions against climate warming. In the present study to enhance the representation of below ground soil processes in ESMs, we replace the CMF that controls the soil OC decomposition in the ORCHIDEE model by the multivariate empirical model established from a data-driven analysis of soil samples by Moyano et al. (2012). This empirical model is an ensemble of HR responses to soil moisture relationships that further accounts for the soil OC content, the soil bulk density and the clay fraction. The modified version of ORCHIDEE embedded with this multivariate empirical model enables doubling the global soil OC stock and improved simulated HR $CO_2$ fluxes in latitudes northern than 40°N. Indeed, variations of $CO_2$ fluxes that occur at short time scale and high frequency, have little influence on litter and soil OC stock on a daily basis. However, after tens of years to centuries any slight change in $CO_2$ flux is significantly driving OC stock variability and therefore the carbon sink potential in models.

To be able to assess the influence of the multivariate empirical model on the carbon cycle scheme in ORCHIDEE, it was more convenient in the present study to employ the bucket C model scheme. It will still be valuable to include the multivariate empirical model in the last generation of ESMs that are structured with C-N cycle interactions and a vertical carbon discretization. Indeed, this combination will benefit in the topsoil of the data-driven approach; meanwhile the N cycle will serve to constrain the NPP and vertical C discretization will favour deeper carbon storage as suggested by Varney et al. (2022).

Accordingly, the present study demonstrates the effectiveness of combining multivariate empirical approach and process-based models to enhance the next generation of ESMs. Recent enrichment of upscaling observation database approaches and of remote sensing products with increasing higher time and space resolutions are strong gestures to carry on this way. Simulating the effect on soil OC storage capacity of repeated short term and high frequency weather events such as extreme drought and precipitation can be challenging for global scale models which could be undertaken by combining process based model and empirical approaches.

## Code availability

The source codes is available online via https://forge.ipsl.jussieu.fr/orchidee/wiki/GroupActivities/CodeAvalaibilityPublication/ORCHIDEE (last access: 17 September 2024). Readers interested in running the model should follow the guidelines at http://orchidee.ipsl.fr/index.php/you-orchidee (last access: 17 September 2024).





**Author contribution**

E.S and B.G. modified the model implementation, performed simulations, contributed to results interpretation and
prepared the manuscript. B.G and A.D. conceptualized, secured funding and supervised the project. All authors contributed
to the interpretation of the results and manuscript revisions.

**Competing interests**

The authors declare that they have no conflict of interest.

**Acknowledgements and Financial support**

This work was funded in part by the Belmont forum BLUEGEM (ANR-21-SOIL-0001) project. E. Salmon
acknowledges funding by the European Horizon project, HORIZON-CL5-2021-D1-01, GreenFeedback (grant agreement
101056921). B. Guenet was supported by the ALAMOD project of the exploratory research program FairCarboN and
received government funding managed by the Agence Nationale de la Recherche under the France 2030 program, reference
ANR-22-PEXF-002-projet ALAMOD.

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
