# Peer review of "Accounting for empirical global soil organic characteristics and moisture heterogeneities in soil organic decomposition scheme of land surface models."

_EGUsphere, 2025_

## Author Comment (AC1)

**RC1**: 'Comment on egusphere-2025-3511', Anonymous Referee #1,

https://doi.org/10.5194/egusphere-2025-3511-RC1

The authors identified the clear lack in current LSMs and ESMs of using monotonically increasing CMF-functions, although a hill-shaped function is closer to reality by modeling anoxic effects. The proposed integration of a multivariate function by Moyano et. al is clearly a mechanistic improvement, which is argumentatively well explained. Anyway could these advances in the evaluation not clearly be supported, which could maybe done with further evaluation. I propose to consider the manuscript for publication in ESD after major revisions addressing the points below. The additional evaluation requested may, however, reveal structural issues or inconsistencies in the results, which could ultimately justify rejection if major flaws are identified.

Thank you for your support on the proposed integration work that is presented here. We understand concerns that the reviewer has pointed out (about possible inconsistencies). However, in the event that the model evaluation could not be fully explained, we believe that the implementation of a more physical based approach is by itself significant enough to be shared with the community. To reassure on the efficiency of the integration of the mechanistic approach we addressed point by point reviewers comments below. Line numbers indicated in the responses to comments are the line of the track changed manuscript.

Major revisions:

1. Assumptions on the effect of wetlands in the evaluation products are plausible, but with masking those and the ORCHIDEE(-M) predictions these assumptions should be tested. As there is overestimation of stocks in tropics by the M-version, this downside should be evaluated further, which can only be done by masking. Afterwards should the actual spatial correlation be additionally plotted.

We fully agree with the reviewer that this assumption must be tested and we are currently working on versions of the model that include wetlands (see Salmon et al., 2022 for an attempt on peatlands or Zhao et al., 2025 for mangroves) but wetlands in general are very complex ecosystems and so far the model does not consider wetlands. Furthermore, we want to underline that the objective of the paper is to show how the global simulated SOC stock can by affected by using a mechanistic approach of the relationship between soil moisture and SOC decomposition rate and not to demonstrate the representation of the wetland SOC stocks which is a whole topic by itself (Hugelius et la. 2014; Lindgren et al., 2018; Minasny et al., 2024). In our manuscript we hypothesize that wetlands SOC stocks might explain the lack of SOC stock which does not prevent the model from being biased since before or after modifying the model no calibration has been performed on purpose to evaluate only the change in the control moisture function. To better emphasize this limitation we will modify the manuscript by adding : Lines 419-425 "It should be pointed out that wetland ecosystems such as northern and tropical peatlands, floodplains and mangroves are not explicitly represented in this version of the model. Wetlands ecosystems are quite complex to represent in LSMs and some progress has been done for ORCHIDEE concerning peatlands (Qiu et al, 2020,

Salmon et al., 2024) and concerning mangroves (Zhao et al., 2025) but so far we do not have a version of the model that consider the wetlands ecosystems in their entire complexity.  Notwithstanding the lack of explicit wetland soil processes, the simulated soil OC stocks by the modified ORCHIDEE version reach 170 Pg at higher latitudinal bands 40°N-80°N and 300 Pg in tropical regions at 40°N-40°S."

Related to the masking, we tried in a previous draft of the manuscript but we faced some methodological flaws. In particular, we are not aware of a global product that gives only the SOC from wetlands. Therefore, we tried to filter out the gridcells of a total SOC product such as Soilgrids that have a high proportion of histosols (assuming that all peatlands are histosols). By doing so, we realized that the results were quite sensitive to the value of the thresholds we decided to filter out the pixels leading us to doubt about the robustness of this approach. Instead, we propose here to compare the surface area i.e. the grid cell of the model at 0.5° by 0.5° of the soil organic carbon stock estimated by the databases GSDE, HWSD and Soilgrids, with the estimation obtained by the ORCHIDEE-M model and the peatlands fraction of the PEATMAP product from Xu et al. (2018). In Figure R1 are displayed in dark blue surface areas having observed soil organic carbon stock that is lower than the amount simulated by ORCHIDEE-M and having a peatland fraction not null. In brown are surface areas with higher soil organic carbon stock in the observation databases than in the model and a positive peatlands fraction. And in green and yellow are the areas with a peatland fraction of zero, respectively, having more soil organic carbon stock in the observation products than in the model and having less soil organic carbon stock in the observation products than in the model. Globally these three maps show larger surface areas where a fraction of peatland is not null and having an underestimation of the soil organic carbon stock by the model ORCHIDEE-M compared to the products derived from observations ( brown areas). Dark blue surface areas that correspond to areas where the model is overestimating the soil organic carbon stock and overlaps with positive peatland fraction have a smaller extent than the brown areas.

We will modify the manuscript by adding the Figure R1 in an appendix and referring to this appendix in the manuscript:

Line 302 "Both versions of the model have been unable to capture the sharper peak of soil OC density in the equatorial band likely originating from tropical wetland ecosystems (Appendix A1). "

[Figure]

Figure R1: Areas colored maps showing overlaps of areas where soil organic carbon stock estimated by the database GSDE, HWSD and SoilGrids are larger or smaller than simulated soil organic carbon stock by the ORCHIDEE-M model and areas having a peatland fraction (higher than zero) in the PEATMAP product (Xu et al., 2018)

Salmon, E., Jégou, F., Guenet, B., Jourdain, L., Qiu, C., Bastrikov, V., Guimbaud, C., Zhu, D., Ciais, P., Peylin, P., Gogo, S., Laggoun-Défarge, F., Aurela, M., Bret-Harte, M. S., Chen, J., Chojnicki, B. H., Chu, H., Edgar, C. W., Euskirchen, E. S., Flanagan, L. B., Fortuniak, K., Holl, D., Klatt, J., Kolle, O., Kowalska, N., Kutzbach, L., Lohila, A., Merbold, L., Pawlak, W., Sachs, T., and Ziemblińska, K.: Assessing methane emissions for northern peatlands in ORCHIDEE-PEAT revision 7020, Geosci. Model Dev., 15, 2813–2838, https://doi.org/10.5194/gmd-15-2813-2022, 2022.

Zhao, Z., Ciais, P., Guenet, B., Stegehuis, A., Lauerwald, R., Regnier, P., ... & Li, W. (2025). ORCHIDEE-MAN: Incorporating mangrove processes in the global vegetation model of ORCHIDEE. *Journal of Advances in Modeling Earth Systems*, *17*(12), e2025MS005185. https://doi.org/10.1029/2025MS005185

Hugelius, G., Strauss, J., Zubrzycki, S., Harden, J. W., Schuur, E. A. G., Ping, C.-L., Schirrmeister, L., Grosse, G., Michaelson, G. J., Koven, C. D., O'Donnell, J. A., Elberling, B., Mishra, U., Camill, P., Yu, Z., Palmtag, J., and Kuhry, P.: Estimated stocks of circumpolar permafrost carbon with quantified uncertainty ranges and identified data gaps, Biogesciences, 11, 6573–6593, https://doi.org/10.5194/bg-11-6573-2014, 2014.

Lindgren, A., Hugelius, G., and Kuhry, P.: Extensive loss of past permafrost carbon but a net accumulation into present-day soils, Nature, 560, 219–222, https://doi.org/10.1038/s41586-018-0371-0, 2018.

Minasny, B., Adetsu, D.V., Aitkenhead, M. et al. Mapping and monitoring peatland conditions from global to field scale. Biogeochemistry 167, 383–425 (2024). https://doi.org/10.1007/s10533-023-01084-1

Xu, J., Morris, P. J., Liu, J., & Holden, J. (2018). PEATMAP: Refining estimates of global peatland distribution based on a meta-analysis. *Catena*, *160*, 134-140. https://doi.org/10.1016/j.catena.2017.09.010

2. All evaluation products are themselves predictions/generalizations of ground truth measurements. Evaluation should additionally be done on the ground truth dataset behind the products.

Usually, models are validated against "ground truth" dataset at site level for optimisation studies for instance. While here we aim at evaluating sensitivity of the model for different formulations of a process therefore we choose as commonly done in other LSM studies (Ito et al., 2020; Varney et al., 2022) to compare the results to data-driven products. We feel like in the present study it is more relevant to compare the simulation results to data-driven products that provide estimates of SOC stocks and HR flux at the grid cell resolution like in our model. Indeed, in this version of the ORCHIDEE model the control moisture function is estimated at the grid cell level consequently soil carbon decomposition is also at the grid cell level and not at the pixel level i.e. the PFT soil tile. Therefore, a comparison of the simulation results with site level SOC stocks as displayed in Figure R2 using the World Soil Information Service (WoSIS) of the International Soil Reference and Information Centre (ISRIC) database, is less significant than using a gridded data-driven product.

Calisto, L., de Sousa, L.M., Batjes, N.H., 2023. Standardised soil profile data for the world (WoSIS snapshot – December 2023), https://doi.org/10.17027/isric-wdcsoils-20231130

Batjes N.H., Calisto, L. and de Sousa L.M., 2023. Providing quality-assessed and standardised soil data to support global mapping and modelling (WoSIS snapshot 2023). Earth System Science Data, https://doi.org/10.5194/essd-16-4735-2024.

[Figure]

Figure R2: Comparison of simulated SOC stock estimated by the ORCHIDEE and the ORCHIDEE-M version against the World Soil Information Service (WoSIS) of the International Soil Reference and Information Centre (ISRIC) database.

3. As ORCHIDEE(-M) produces subdaily predictions and the effect of moisture on HR can assumed to be a very dynamic one, additional evaluation with subdaily products as (e.g.) Fluxcom Reco could reveal additional insights and integrate additionally tower measurements into evaluation.

We agree with the reviewer that a full detailed comparison of the model results against site observations will be very interesting. However, such study should be performed on a calibrated version of the model. Our objective for this manuscript is to promote a mechanistic approach of the relationship between soil moisture, soil carbon content and clay content and in any case to demonstrate the efficiency of a model calibration.

4. The M-version is supposed to model also anoxic conditions. It should be further elucidated, why wetlands are assumed to still be missed.

Indeed but here the limitation is not related to the effect of anoxic conditions on decomposition but more about the ability of the model to reproduce saturated soil conditions. Floodability of wetlands ecosystems rely on topography (they are usually located in lowland) that convey some of the runoff flow to the wetland area, the precipitation and upward movement of the groundwater flow while hydrological processes in ORCHIDEE rely only on the precipitation. In this version, the hydrology of the model is not able to represent saturated soil. In other ORCHIDEE versions such as the ORCHIDEE-PEAT version (Qiu et al. 2019), to increase the soil moisture content in the peatland soil tile, we have to add subgrid lateral flow of runoff to the peatland soil tile. Also the high SOC content and the large porosity is changing thermal conductivity and heat capacity of the peatland soil tile. Parametrization of the peatland phenology also needs to be adapted to part time oxic and part time anoxic conditions.

Qiu, C., Zhu, D., Ciais, P., Guenet, B., Peng, S., Krinner, G., Tootchi, A., Ducharne, A., and Hastie, A.: Modelling northern peatland area and carbon dynamics since the Holocene with the ORCHIDEE-PEAT land surface model (SVN r5488), Geosci. Model Dev., 12, 2961–2982, https://doi.org/10.5194/gmd-12-2961-2019, 2019.

Minor revisions:

1. 15: actually there are additional HR modifiers.

   We modified the sentence lines 15-17 : "Current soil HR modifiers employed in models mainly are a single relationship between soil moisture and the rate of decomposition that are employed for all the ecosystem types."

2. 48: residuals?

   We modified the sentence lines 49-50 : "Guenet et al., (2024) showed that precipitation is a key driver of the HR ESMs' residues suggesting that a better representation of the soil moisture effect on decomposition may be a good lead to improve HR representation in ESMs."

3. 51: I would say the moisture product forces, while constraining would mean a traget used to optimized parameters.

   We modified the sentence lines 51-52 : "Furthermore, the uncertainty of HR data-driven estimates is also widely dependent on the soil moisture product employed to force the database "

4. 58: Q10 formulation itself is not hill-shaped and conceptually the same as Arrhenius.

We agree with the review therefore we rephrase this long sentence and the references as below:

Lines 55-61: "Varney et al., (2022) investigated variability in soil OC stocks estimated from ESMs involved in the Coupled Model Intercomparison Project (CMIP), CMIP5 and CMIP6, and distinguished two three types of temperature schemes and two moisture schemes. The temperature schemes are two : (1) an increase relationships using Arrhenius law or Q10 formulation for temperature and a hill-shape relationship such as in the BCC-CSM2-MR (Wu et al., 2019) and GFDL-ESM models (Dunne et al., 2020). The moisture schemes are a monotonically increasing function with increasing soil moisture and a hill function that increase to an optimum moisture level then decrease. "

Dunne, J. P., John, J. G., Shevliakova, E., Stouffer, R. J., Krasting, J. P., Malyshev, S. L., Milly, P., Sentman, L. T., Adcroft, A. J., Cooke, W., Dunne, K. A., Griffies, S. M., Hallberg, R. W.,

Harrison, M. J., Levy, H., Wittenberg, A. T., Phillips, P. J., and Zadeh, N.: GFDL's ESM2 global coupled climate–carbon earth system models. Part II: carbon system formulation and baseline simulation characteristics, J. Climate, 26, 2247–2267, 2013.

Wu, T., Lu, Y., Fang, Y., Xin, X., Li, L., Li, W., Jie, W., Zhang, J., Liu, Y., Zhang, L., Zhang, F., Zhang, Y., Wu, F., Li, J., Chu, M., Wang, Z., Shi, X., Liu, X., Wei, M., Huang, A., Zhang, Y., and Liu, X.: The Beijing Climate Center Climate System Model (BCC-CSM): the main progress from CMIP5 to CMIP6 , Geosci. Model Dev., 12, 1573—1600, https://doi.org/10.5194/gmd-12-1573-2019, 2019.

5. 60-65: I had real difficulties in understanding this paragraph's message.

We added a sentence at the end of this paragraph to explain the message.

Lines 62-70: "Moreover, Falloon et al. (2011) appraised soil carbon changes responsiveness to control moisture- respiration functions embedding twelve representative climate models' functions in the RothC model. These functions have various shapes but provide the lowest of rate modifier values at the lowest soil moisture content and, for half of the functions, a rate modifier that is maximum at saturation. For the other half, a maximum rate near or higher than 0.5 is given at the saturation fraction.  They aimed at defining the responsive range of control moisture- respiration functions on the global soil moisture on soil carbon changes between 1860 and 2100 and showed that soil carbon changes in a range of -2 to +3% between 1860 and 2100 depending on the control moisture- respiration function chosen. This demonstrates models' sensitivity of the soil moisture- respiration functions on soil OC accumulation."

6. 68: Linear means here factor of 1? Why do ratios lower than 1 indicate non-linear relationships?

We agree with the reviewer a lower ratio value does not no linear relationship therefore will modify this sentence :

Lines 74-78: "Conversely, the latest regional data-driven HR estimate by Ciais et al. (2020) suggests otherwise. Indeed, the HR to net primary productivity (NPP) ratios of 9 large regions are lower than 1 and the average ratios of each region range between 0.37 and 0.85 suggesting a non linear. The authors suggested that carbon lateral transfer to aquatic ecosystems that is rarely accounted for in ecosystem models that participate in CMIP work could influence the relationship between the C input supply by litterfall and $CO_2$ emissions resulting from HR, driven by carbon lateral transfer to aquatic ecosystems.  "

7. 77-79: I did not understand the meaning of the last part of the sentence.

We modified this last part line 79-87 such as: "Other conditions such as soil pore space, bulk density and texture are neglected in process-based models regardless of their influence on metabolic activities of aerobic organisms (Moyano et al. 2013). Moyano et al. (2012) provide an analysis of the soil

moisture response on HR of a wide range of mineral soil types and organic-richer soils. From this database, they provided a multivariable model dependent on soil moisture and soil characteristics namely soil organic carbon content, clay fraction and bulk density. Thus, this empirical model is not using a single set of parameters to link soil moisture and SOC decomposition at any location  but an ensemble of parameters that depends on local conditions  enabling to consider spatial heterogeneity. Regardless of its meaningful quality, this empirically-based model has never been employed in a process-based model."

8. 90: I would have needed a short introduction into the parameter PRSR.

A short introduction has been added Lines 97-102:

"Moyano et al. (2012) established empirical models accounting for the dependence of the Proportional Response of Soil Respiration (PRSR) with soil characteristics. The authors assumed that HR varies with the response to changes in soil moisture and established a relationship of the proportion of HR changes, named PRSR, related to a 0.01 increase in soil moisture. Then an analysis of this relationship between PRSR and SM for different properties of soil served to formulate using a generalized additive model (Hastie and Tibshirani, 1986 and 1987), three predicting formulations of the PRSR for mineral soils and organic rich soil accounting for soil bulk density, clay content and SOC content."

Hastie, T., & Tibshirani, R. (1986). Generalized additive models. Statistical science, 1(3), 297-310.

Hastie, T., & Tibshirani, R. (1987). Generalized additive models: some applications. Journal of the American Statistical Association, 82(398), 371-386.

9. 91-93: I think the order of the sentence is not correct.

We modified this sentence Lines 102-105:

"We  use the land surface model ORCHIDEE v2.2 revision 8416 (Boucher et al., 2020) , named hereafter ORCHIDEE,  to evaluate changes involved on simulated HR et SOC content when using these empirical models . "

10. 100, 114: Texture is a control in the standard version already. It would be interesting to discuss why this is not enough.

This is indeed a good point. The standard texture control is mimicking the interactions between SOC and minerals and how these interactions may affect the decomposition whereas in Moyano et al. approach, the texture is driving the response SOC decomposition on soil moisture changes. We added these sentences in the text: lines 144-159 "The clay fraction is defined using Zobler (1986),  and the Harmonized World

Soil Database (HWSD) soil map (Fischer et al., 2008) is used to define soil bulk density values of each grid cell. It is important to note that the effect of texture described in eq. 1 and eq. 4 are not equivalent. In eq. 1, $f(\gamma)$ is simulating the interactions between mineral matrix and the SOC leading to potentially increase the SOC stability whereas in eq.4, the $\beta 4\ clay$ terms is driving the effect of soil moisture on decomposition that is affected by the clay content."

11. 109: The moisture could be below 25% but the relationship is constant below 25%.

Soil moisture can be below 25%, what this means is that when soil is very dry, the decomposition cannot be less than 25% of the potential.

12. 123-124: I could not identify where the linear regression models are coming from.

Following comment 8 above, we have added a short introduction to section 2.1 Model description that briefly explains how Moyano et al. 2012 build their model using general additive model approach. To be consistent and accurate we modified the sentence below:

Lines 150-152: "In order to  use the empirical model within its validation space, environmental constraints, i.e. clay fraction, soil OC and the bulk density, are retained within ranges defined by Moyano et al. (2012) based on the soil samples used to fit  the general additive models. "

13. 134: Confusion about the indices n=1 in brackets and then n-1=0.

We modify n-1=0 to n-1 to be less confusing for the reader, the sentence read now line 162:

"For the first interval number (n=1), the value of $MPR_{SR}(\theta_s)_{n-1}$ is 1. "

14. 135: The mentioned subtraction is not visible in the function above.

Indeed, the subtraction is not in equation 5 because it is the next step applied to MPR$_{SR}$ that is explained in this sentence: lines 162-164 "The soil moisture control function (MPR$_{SR}$) is scaled to range between 0 and 1 by subtracting all values with the smallest value then normalizing with the highest one. " and not described in an equation formalisme.

15. Fig. 1: Organic rich is difficult to distinguish from mineral soil values. Furthermore would it be interesting to see the effects of clay vs. OC in the mineral function.

We modify to solid black the line color for the organic rich values. The effect of clay, OC and bulk density on the shape of the function is displayed in Figure 3 of Moyano et al., (2012) so we did not want to duplicate this figure or a subpart of it. However we showed in the result section Fig. 3 the relationship between clay, OC and the value of the control function used in both model versions.

[Figure]

**Figure 1: Representation of the control moisture function in ORCHIDEE (dash-dotted line, Krinner et al., 2005) and of the ensemble of control moisture function embedded in the modified ORCHIDEE-M version for the mineral soil (light grey solid lines) using clay fractions ranging between 0.27 and 0.34, soil OC ranging between 0.01 and 0.06 g C / g soil and for organic-richer soil (black solid line) with soil OC higher than 0.06 g C / g soil.**

16. 161: If SoilGrids v2.0 was used, then Poggio not Hengl was first author. If not, why was not v2.0 used? Does SoilGrids (v2.0) claim to model peatlands correct?

Thank you for pointing that out, we indeed used SoilGrids (v2.0) and modified the reference in the modified manuscript.

Lines 187-190: "To evaluate the model, a global-scale benchmarking of simulated carbon stock was performed using three soil datasets estimating soil OC stock from 0 to 1m namely the Global Soil Dataset for use in Earth System Models (GSDE- Shangguan et al., 2014), Harmonized World Soil Database version 2.0 (HWSD v2.0 - FAO & IIASA, 2023) and the global gridded soil system SoilGrids v2.0 (Poggio et al., 2021). "

I am not sure whether or not SoilGrid (v2.0) models peatlands. Poggio et al. (2021) does not mention peatlands and claimed that "Soil property data for this study were derived from the ISRIC World Soil Information Service (WoSIS), which provides consistent, standardised soil profile data for the world (Batjes et al., 2020)." Then Batjes et al. (2020) does not mention peatlands and explain "The ISRIC-WISE profile database (Batjes, 2009, 2011) was complemented with some 8000 'new' profiles, originating mainly from North America (ISCN, 2014) and 'High Latitude' regions (Harden et al., 2012; Hugelius et al., 2014; Michaelson et al., 2013)." Harden et al., 2012; Hugelius et al., 2014; Michaelson et al., 2013 are prior studies to Hugelius et

al. 2020 database on peatlands and permafrost SOC stocks in which the authors estimated an average SOC density of 115 kg m$^{-2}$ and an average soil depth of 249 cm.

Batjes, N. H., Ribeiro, E., and van Oostrum, A.: Standardised soil profile data to support global mapping and modelling (WoSIS snapshot 2019), Earth Syst. Sci. Data, 12, 299–320, https://doi.org/10.5194/essd-12-299-2020, 2020.

Harden, J.W., Koven, C.D., Ping, C.-L., Hugelius, G., David McGuire, A., Camill, P., Jorgenson, T., Kuhry, P., Michaelson, G.J., O'Donnell, J.A., Schuur, E.A.G., Tarnocai, C., Johnson, K., Grosse, G., 2012. Field information links permafrost carbon to physical vulnerabilities of thawing. Geophys. Res. Lett. 39 (15), L15704.

Hugelius, G., Strauss, J., Zubrzycki, S., Harden, J.W., Schuur, E.A.G., Ping, C.-L., Schirrmeister,L., Grosse, G., Michaelson, G.J., Koven, C.D., O'Donnell, J.A., Elberling, B., Mishra, U.,Camill, P., Yu, Z., Palmtag, J., Kuhry, P., 2014. Estimated stocks of circumpolar permafrost carbon with quantified uncertainty ranges and identified data gaps. Biogeosciences 11, 6573–6593.

Michaelson, G.J., Ping, C.-L., Clark, M., 2013. Soil pedon carbon and nitrogen data for Alaska: an analysis and update. Open J. Soil Sci. 3, 132–142.

Hugelius, G., Loisel, J., Chadburn, S., Jackson, R. B., Jones, M., MacDonald, G., ... & Yu, Z. (2020). Large stocks of peatland carbon and nitrogen are vulnerable to permafrost thaw. *Proceedings of the National Academy of Sciences*, *117*(34), 20438-20446. https://doi.org/10.1073/pnas.1916387117

17. 183: distribution not dispersal.

We modified the sentence lines 211-213: "Differences in the CMF determined in both model versions, ORCHIDEE and ORCHIDEE-M for the litter (on the left side) and the soil (on the right side) are displayed in Fig. 2. The bar plots (A to D) show the distribution of grid cells number for various CMF ranges of values. "

18. 186: The description of CMF strength is confusing; low CMF implies strong limitation on HR.

CMF ranges between 0 and 1 modifying the rate of decomposition (see equation 1: $k_p \cdot [C]_p \cdot f(\theta)$ ) therefore if the rate is multiplied by 1, CMF has no limitation on the rate of decomposition and a low value of CMF 0.2 will significantly reduce the rate. We will modify this sentence to:

lines 214-216: "In the standard model version, for the litter and the soil, 37% and 54% of the grid cells respectively have CMF values higher than 0.9 meaning that litter and soil moisture content were ideal for decomposition leading to high decomposition rates has no significant influence on the HR. "

19. 193: 'negative values lead to higher values' is a little counterintuitive.

We modified this sentence:

Lines 222 -223: "In Fig. 2E and 2F maps, green and blue areas  indicate that CMF values are higher in the standard version, enabling a faster soil OC decomposition. "

20. Fig. 2: Mention that mean annual values are indicated here.

We modified the figure caption lines 244-247 "Figure 2: At the top, bar charts of the number of grid cells per control moisture function value ranges for the litter (A) and the soil (C) in the ORCHIDEE version, for the litter (B) and the soil (D) in the ORCHIDEE-M version. The bottom rows are maps of values difference between the annual mean control moisture function in the ORCHIDEE and ORCHIDEE-M versions for litter (E) and for soil (F)."

This sentence we modify "OC stock rate" to "OC stock":

Lines 269 - 270: "In both model versions, the OC stock  depends on the OC decomposition rate, for the litter, on the material originating from the net primary productivity and for the soil OC stock, on the input of the litter OC content. "

22. 308: Table 2 & 3 would be much clearer if PFT's would directly be written out. Abbreviations are not necessary.

We modified the tables as below:

**Table 2: Simulated litter and soil organic carbon content from both model versions ORCHIDEE (ORC) and ORCHIDEE-M (ORC-M) and for the ecosystems considered in the model. The organic carbon content differences of ORCHIDEE minus ORCHIDEE-M version is provided in petagram and in percent of the ORCHIDEE OC content estimated for each plant functional type (PFT).** ~~which are TrBLE for Tropical Broad-leaved Evergreen, TrBLR for Tropical Broad-leaved Raingreen, TeNLE for Temperate Needle-leaf Evergreen, TeBLE for Temperate Broad-leaved Evergreen, TeBLS for Temperate Broad-leaved Summergreen, BNLE for Boreal Needle-leaf Evergreen, BBLS for Boreal Broad-leaf Summergreen, BNLS for Boreal Needle-leaf Summergreen, TeC3G for Temperate C3 Grass, C4G for C4 dominated grassland, C3Agri for C3 Agricultural plants, C4Agri for C4 Agricultural plants, TrC3G for Tropical C3 dominated Grassland and BC3G for Boreal C3 dominated Grassland.~~ **Fractions of PFTs indicate the percentage of total area of each PFT of the total land surface.**

| | | Litter | | | | Soil | | | |
|---|---|---|---|---|---|---|---|---|---|
| Plant Functional Types | Fraction | ORC | ORC-M | Differences | | ORC | ORC-M | Differences | |
| | % | Pg | Pg | Pg | %C | Pg | Pg | Pg | %C |
| Tropical Broad-leaved Evergreen | 9.4% | 10.6 | 13.8 | 3.3 | 31% | 59.8 | 75.3 | 15.5 | 26% |
| Tropical Broad-leaved Raingreen | 6.0% | 5.2 | 9.1 | 3.9 | 75% | 27.5 | 47.0 | 19.5 | 71% |
| Tropical C3 dominated Grassland | 3.3% | 0.9 | 1.4 | 0.5 | 57% | 5.0 | 7.8 | 2.8 | 57% |
| C4 dominated grassland | 3.2% | 16.1 | 31.3 | 15.2 | 94% | 75.5 | 138.3 | 62.8 | 83% |
| Temperate Needle-leaf Evergreen | 3.5% | 1.6 | 2.6 | 1.0 | 61% | 9.1 | 14.2 | 5.1 | 55% |
| Temperate Broad-leaved Evergreen | 4.8% | 4.0 | 7.0 | 3.0 | 75% | 23.0 | 38.0 | 15.1 | 66% |
| Temperate Broad-leaved Summergreen | 3.8% | 3.6 | 6.4 | 2.8 | 78% | 20.4 | 33.7 | 13.3 | 65% |
| Temperate C3 Grass | 2.7% | 4.6 | 8.2 | 3.6 | 78% | 25.3 | 43.4 | 18.1 | 72% |
| C3 Agricultural plants | 6.8% | 8.1 | 14.3 | 6.2 | 77% | 35.9 | 62.6 | 26.6 | 74% |
| C4 Agricultural plants | 9.0% | 3.6 | 6.5 | 2.9 | 80% | 12.1 | 20.9 | 8.8 | 73% |

| | | | | | | | | | |
|---|---|---|---|---|---|---|---|---|---|
| Boreal Needle-leaf Evergreen | 9.1% | 3.8 | 7.2 | 3.4 | 88% | 19.1 | 35.1 | 16.0 | 84% |
| Boreal Broad-leaf Summergreen | 2.2% | 3.5 | 6.7 | 3.2 | 89% | 19.1 | 35.3 | 16.1 | 84% |
| Boreal Needle-leaf Summergreen | 4.1% | 1.4 | 2.8 | 1.5 | 108% | 8.6 | 18.5 | 9.9 | 115% |
| Boreal C3 dominated Grassland | 11.4% | 6.0 | 11.3 | 5.3 | 89% | 32.1 | 60.6 | 28.5 | 89% |

**Table 3: Simulated heterotrophic respiration $CO_2$ fluxes from both model versions ORCHIDEE and ORCHIDEE-M. The CO2 flux differences of ORCHIDEE minus ORCHIDEE-M version is provided in teragram (Tg) and in percent of the ORCHIDEE CO2 fluxes estimated for each plant functional type (PFT).** ~~which are TrBLE for Tropical Broad-leaved Evergreen, TrBLR for Tropical Broad-leaved Raingreen, TeNLE for Temperate Needle-leaf Evergreen, TeBLE for Temperate Broad-leaved Evergreen, TeBLS for Temperate Broad-leaved Summegreen, BNLE for Boreal Needle-leaf Evergreen, BBLS for Boreal Broad-leaf Summegreen, BNLS for Boreal Needle-leaf Summergreen, TeC3G for Temperate C3 Grass, C4G for C4 dominated grassland, C3Agri for C3 Agricultural plants, C4Agri for C4 Agricultural plants, TrC3G for Tropical C3 dominated Grassland and BC3G for Boreal C3 dominated Grassland.~~

| Plant Functional Types | ORCHIDEE | ORCHIDEE-M | Differences | |
|---|---|---|---|---|
| | Tg / yr | Tg / yr | Tg / yr | %C |
| Tropical Broad-leaved Evergreen | 20.14 | 20.11 | -0.028 | -0.14% |
| Tropical Broad-leaved Raingreen | 8.64 | 8.60 | -0.033 | -0.39% |

| | | | |
|---|---|---|---|
| Tropical C3 dominated Grassland | 1.51 | 1.51 | 0.000 | 0.01% |
| C4 dominated grassland | 21.05 | 20.82 | -0.230 | -1.09% |
| Temperate Needle-leaf Evergreen | 1.87 | 1.86 | -0.014 | -0.72% |
| Temperate Broad-leaved Evergreen | 3.45 | 3.44 | -0.013 | -0.38% |
| Temperate Broad-leaved Summergreen | 4.13 | 4.08 | -0.051 | -1.22% |
| Temperate C3 Grass | 4.81 | 4.75 | -0.058 | -1.21% |
| C3 Agricultural plants | 8.28 | 8.20 | -0.083 | -1.00% |
| C4 Agricultural plants | 4.06 | 4.01 | -0.045 | -1.12% |
| Boreal Needle-leaf Evergreen | 2.34 | 2.31 | -0.030 | -1.28% |
| Boreal Broad-leaf Summergreen | 2.34 | 2.31 | -0.026 | -1.12% |
| Boreal Needle-leaf Summergreen | 0.67 | 0.67 | -0.001 | -0.14% |
| Boreal C3 dominated Grassland | 3.19 | 3.15 | -0.038 | -1.19% |